# Gender Differences in the Impact of Plasma Xanthine Oxidoreductase Activity on Coronary Artery Spasm

**DOI:** 10.3390/jcm10235550

**Published:** 2021-11-26

**Authors:** Ken Watanabe, Tetsu Watanabe, Yoichiro Otaki, Takayo Murase, Takashi Nakamura, Shigehiko Kato, Harutoshi Tamura, Satoshi Nishiyama, Hiroki Takahashi, Takanori Arimoto, Masafumi Watanabe

**Affiliations:** 1Department of Cardiology, Pulmonology and Nephrology, Yamagata University School of Medicine, Yamagata 990-9585, Japan; k.watanabe0418@med.id.yamagata-u.ac.jp (K.W.); y-otaki@med.id.yamagata-u.ac.jp (Y.O.); sg-kato@med.id.yamagata-u.ac.jp (S.K.); htamura@med.id.yamagata-u.ac.jp (H.T.); mnisiyam@med.id.yamagata-u.ac.jp (S.N.); hitakaha@med.id.yamagata-u.ac.jp (H.T.); t-arimoto@med.id.yamagata-u.ac.jp (T.A.); m-watanabe@med.id.yamagata-u.ac.jp (M.W.); 2Radioisotope and Chemical Analysis Center, Sanwa Kagaku Kenkyusho Co., Ltd., Inabe 511-0406, Japan; ta_murase@skk-net.com; 3Pharmaceutical Research Laboratories, Pharmacological Study Group, Sanwa Kagaku Kenkyusho Co., Ltd., Inabe 511-0406, Japan; ta_nakamura@mb4.skk-net.com

**Keywords:** xanthine oxidoreductase, coronary artery spasm, gender differences

## Abstract

Xanthine oxidoreductase (XOR) is the rate-limiting enzyme in uric acid (UA) production that plays a pivotal role in generating oxidative stress. Gender differences in the impact of plasma XOR activity on coronary artery spasm (CAS) remain unclear. We investigated plasma XOR activity in 132 patients suspected of having CAS (male, *n* = 78; female, *n* = 54) and who underwent an intracoronary acetylcholine provocation test. Plasma XOR activity was significantly lower in female patients compared with male patients. CAS was provoked in 36 male patients and 17 female patients, and both had significantly higher plasma XOR activity than those without. Multivariate logistic regression analysis showed that this activity was independently associated with the incidence of CAS in both sexes after adjusting for confounding factors. The optimal cut-off values for predicting CAS were lower in female patients than in male patients. Multivariate analysis demonstrated that female patients with high XOR activity exhibited a higher incidence of CAS than male patients. Plasma XOR activity was an independent predictor of the incidence of CAS in both sexes. The impact of plasma XOR activity on CAS was stronger in female patients than in male patients.

## 1. Introduction

Coronary artery spasm (CAS) is an important cause of acute coronary syndrome (ACS) and sudden death [1]. Patients with CAS are associated with poor prognosis compared with those without CAS in ACS patients [2]. It has been reported that women have higher mortality rates than men after myocardial infarction [3]. It was reported that female patients with CAS had more frequently diffuse spasm by acetylcholine tests than male patients [4].

Decreased nitric oxide (NO) bioavailability due to increased reactive oxygen species (ROS) is one of the most important causes of CAS [5]. Uric acid (UA) is the end-product of purine metabolism that can induce inflammation and ROS production in vascular endothelial cells, leading to a number of cardiovascular diseases [6,7]. It has been demonstrated that serum UA is independently correlated with CAS [8].

Xanthine oxidoreductase (XOR) is a pivotal enzyme in the production of UA that is accompanied by the generation of ROS [9]. Increased levels of XOR have been recognized as a high risk factor for cardiovascular diseases, such as heart failure and coronary artery disease, including CAS [10,11,12,13]. It is well known that gender differences exist in the impact of serum UA levels on cardiovascular risk [14]. However, little is known about the gender differences in plasma XOR activity. The aim of this study, therefore, was to investigate gender differences in the impact of plasma XOR activity on CAS.

## 2. Materials and Methods

### 2.1. Study Subjects

We investigated plasma XOR activity in 132 patients (male, *n* = 78; female, *n* = 54) suspected of having CAS due to episodes of chest pain that occurred during rest, not exertion, in the early morning or late at night. All patients underwent an intracoronary acetylcholine provocation test in our hospital between June 2008 and October 2016. Intracoronary infusion of acetylcholine was performed according to the CAS guidelines of the Japanese Circulation Society [15]. Before performing the acetylcholine test, we obtained the control coronary angiography. Acetylcholine was injected into the right coronary artery at a dose of 20 or 50 µg and into the left coronary artery at a dose of 20, 50, or 100 µg each over a period of 20 s. Provoked CAS was defined as total or subtotal occlusion (≥90%) with accompanying symptoms of chest pain and/or ischemic ST-segment changes on the electrocardiogram. Vasoactive medications, including calcium channel blockers, nitrates, nicorandil, and other vasodilators, were withdrawn for at least three days before initiating the study. We excluded patients who had significant coronary artery stenosis (≥50%) and/or were taking XOR inhibitors. The diagnoses of hypertension, dyslipidemia, and diabetes mellitus were based on medical records or history of medical therapy. Smoking included both current and past smokers. Clinical data, including age, sex, and medications at discharge, were obtained from medical records. The study protocol was approved by the Institutional Ethics Committee of Yamagata University School of Medicine, and all patients provided written informed consent.

### 2.2. XOR Activity Assay

Blood samples were collected in the early morning within 24 h after admission. Following centrifugation at 3000× *g* for 15 min at 4 °C, and the obtained plasma was stored at −80 °C until analysis. The XOR activity assay was performed using stable isotope-labeled substrate and liquid chromatography-triple quadrupole mass spectrometry (Sanwa Kagaku Kenkyusho Co., Ltd., Nagoya, Japan) [16].

Other biochemistry parameters were measured using routine laboratory methods. The estimated glomerular filtration rate (GFR) was calculated by using the Japanese equation, as previously reported [17].

### 2.3. Statistical Analysis

The results are expressed as the mean ± standard deviation for continuous variables and percentages for categorical variables. Skewed values are presented as median and interquartile range (IQR). Correlations between plasma XOR activity, age, body mass index (BMI), and UA were analyzed using a single linear regression analysis. We used *t*-tests and chi-squared tests to compare continuous and categorical variables, respectively. If the data were not normally distributed, the Mann–Whitney *U*-test was employed. Logistic regression analysis was performed to determine variables independently associated with CAS. Multivariate analysis using a forward stepwise multiple regression model was performed to identify the independent predictors of CAS. Receiver-operating characteristic (ROC) curves for plasma XOR activity were constructed to determine the optimal cut-off values for sensitivity and specificity. Statistical significance was set at *p* < 0.05. All statistical analyses were performed using a standard software package (JMP version 12; SAS Institute, Cary, NC, USA).

## 3. Results

### 3.1. Comparisons of Clinical Characteristics between Males and Females

A comparison of clinical characteristics between male and female patients is shown in Table 1. As seen from the table, male patients were significantly younger, had higher rates of smoking, and higher levels of triglycerides and lower levels of high-density lipoprotein cholesterol (HDL-C) than the female patients. Serum UA levels and plasma XOR activity were significantly lower in female patients than in male patients. Gender differences in the distribution of plasma XOR activity are shown in Figure 1. There were no significant differences in BMI, medication use, prevalence of hypertension, dyslipidemia, and diabetes mellitus between male and female patients. There was a negative correlation between plasma XOR activity and age, and a positive correlation between plasma XOR activity and BMI in male patients. However, there was no correlation between plasma XOR activity, age, and BMI in female patients. In both sexes, there was no significant correlation between plasma XOR activity and levels of serum UA (Figure 2).

### 3.2. Gender Differences in the Impact of Plasma XOR Activity on CAS

CAS was provoked in 36 male and 17 female patients. In both sexes, patients with CAS had significantly higher plasma XOR activity than those without CAS (Figure 3). Univariate and multivariate logistic regression analyses were performed to determine the factors that predict the incidence of CAS. In male patients, multivariate logistic regression analysis showed that plasma XOR activity was independently associated with the incidence of CAS after adjustment for HDL-C and high-sensitivity C-reactive protein (Table 2). Similarly, in female patients, plasma XOR activity was significantly associated with the incidence of CAS after adjustment for age and smoking (Table 3).

Since the results of this study indicated that there were gender differences in plasma XOR activity, we performed ROC analysis to evaluate the best cut-off value for predicting CAS in each sex. As shown in Figure 4, the ROC analysis demonstrated that plasma XOR activity of 91.6 pmol/h/mL was the threshold value for predicting the incidence of CAS in male patients. The ROC analysis also revealed that plasma XOR activity of 52.3 pmol/h/mL was the threshold value for predicting the incidence of CAS in female patients, which was lower than that in male patients. On the other hand, as shown in Figure 5, multivariate analysis demonstrated that female patients with high XOR activity (≥52.3 pmol/h/mL; OR 22.6, *p* < 0.001) exhibited a higher incidence of CAS than male patients (≥91.6 pmol/h/mL; OR 8.2, *p* < 0.001).

Among patients without provoked CAS, there were four patients (male, *n* = 3; female, *n* = 1) with typical chest pain and/or ischemic electrocardiogram changes who might develop a coronary microvascular spasm. Three male patients and one female patient had low XOR activity according to the ROC curve analysis.

## 4. Discussion

The main findings of the present study were as follows: (1) there was a gender difference in the distribution of plasma XOR activity, (2) the optimal cut-off values for predicting CAS were lower in women than in men, (3) high plasma XOR activity was an independent predictive factor for the incidence of CAS in both sexes, and (4) high plasma XOR activity was largely associated with the incidence of CAS in female patients than in male patients.

In the present study, although plasma XOR activity was significantly lower in female patients than in male patients, there was a stronger association between increased plasma XOR activity and the incidence of CAS in female patients. Although the mechanisms of CAS are multifactorial, it has been documented that genetic risk, gene–environment interactions, and mutations in the endothelial nitric oxide synthase (eNOS) gene contribute to CAS, especially in female patients [18,19]. These results indicate that eNOS malfunction is associated with CAS in female patients rather than in male patients. In endothelial cells, eNOS oxidizes L-arginine to L-citrulline and NO, which plays an important role in blood vessel relaxation. XOR-derived ROS can inactivate NO and contribute to eNOS uncoupling. Once uncoupled, eNOS itself generates ROS at the expense of NO, leading to endothelial dysfunction [20,21]. Therefore, it is possible that XOR-derived ROS mediated eNOS downregulation and might affect the high rates of CAS in female patients rather than in males.

Although there is no established consensus on gender differences in plasma XOR activity, Furuhashi et al. reported that males had significantly higher plasma XOR activity than females [22]. Consistent with this report, we observed significantly higher levels of plasma XOR activity in male patients in the present study. Furthermore, it has been reported that plasma XOR activity is correlated with metabolic parameters, insulin resistance, and levels of liver enzymes and adipokines [23]. Adipose tissue is one of the major sources of XOR, which is particularly enhanced in visceral fat in obesity [24]. Males were found to have more visceral adipose tissue, whereas females had more subcutaneous adipose tissue. Sex differences in visceral and subcutaneous fat distribution can possibly explain the positive correlation between plasma XOR activity and BMI in male patients but not in female patients. In addition, differences in sex hormones, including estrogen, may contribute to reduced insulin resistance in female patients [25]. These reports support the results of the present study, in which plasma XOR activity differed between genders.

In the present study, univariate and multivariate logistic regression analyses showed that elderly female patients tended to have a higher risk of CAS. It has been reported that women have lower UA levels because of the uricosuric effect of estrogen [26]. On the other hand, postmenopausal women are at risk of elevated UA levels and cardiovascular disease [27]. However, there is limited information on the association between sex hormones and plasma XOR activity. Considering that visceral fat mass is increased in postmenopausal women [28], elderly female patients could have higher plasma XOR activity, which can contribute to the incidence of CAS. In the present study, despite male patients having a negative correlation between age and plasma XOR activity (R = –0.293, *p* = 0.009), there was no significant correlation between them in female patients (R = –0.097, *p* = 0.487). Although visceral fat mass usually decreases with aging, elderly female patients might have more stored visceral fat, leading to relatively higher levels of plasma XOR activity compared to elderly male patients.

To our knowledge, this is the first study to investigate gender differences in the impact of plasma XOR activity on CAS. Our results suggest that plasma XOR activity is more associated with the incidence of CAS in women than in men. Decreasing XOR activity could be a novel therapeutic target for CAS, especially in female patients. Further studies are needed to examine whether XOR inhibitors are effective for the treatment of CAS.

The current study had several limitations. First, since this was an observational study, the causal relationship between plasma XOR activity and CAS and its impact on gender differences could not be assessed. Second, as we enrolled patients who were suspected of having CAS, gender differences in plasma XOR activity could not be generalized. Finally, because this study enrolled only patients from Japan from a single center, the results might have been affected due to racial bias.

## 5. Conclusions

Plasma XOR activity was an independent predictor of CAS incidence in both sexes. The impact of plasma XOR activity on CAS was stronger in female patients than in male patients.

## Figures and Tables

**Figure 1 jcm-10-05550-f001:**
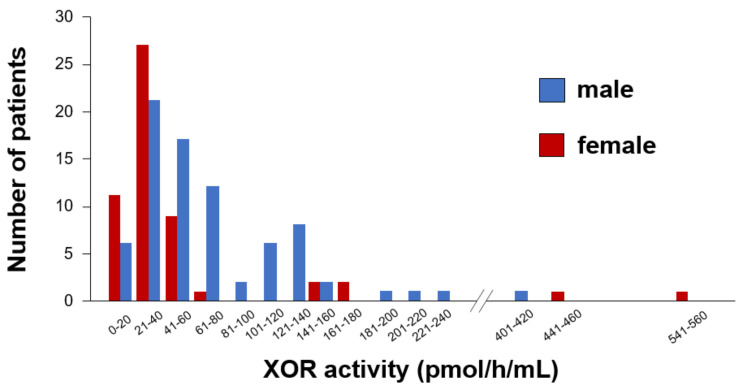
Gender differences in the distribution of plasma XOR activity.

**Figure 2 jcm-10-05550-f002:**
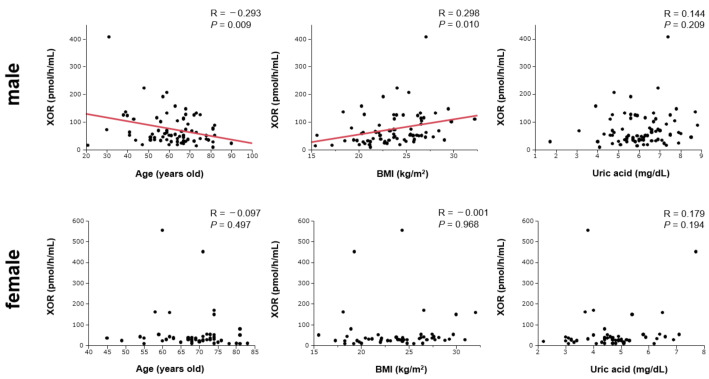
Correlations between plasma XOR activity, age, BMI, and serum UA levels in male and female patients.

**Figure 3 jcm-10-05550-f003:**
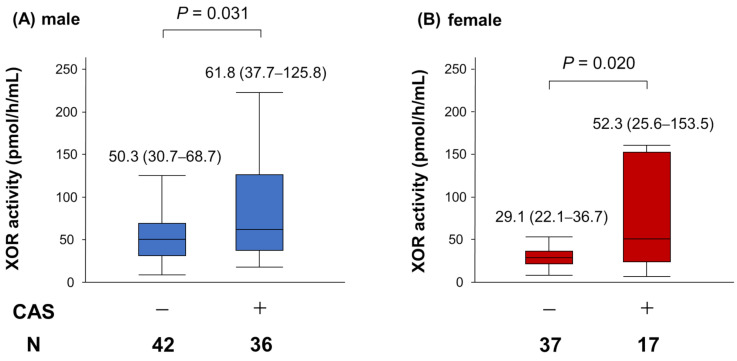
Gender differences in the impact of plasma XOR activity on CAS. (**A**) The comparison of plasma XOR activity between male patients with and without CAS. (**B**) The comparison of plasma XOR activity between female patients with and without CAS.

**Figure 4 jcm-10-05550-f004:**
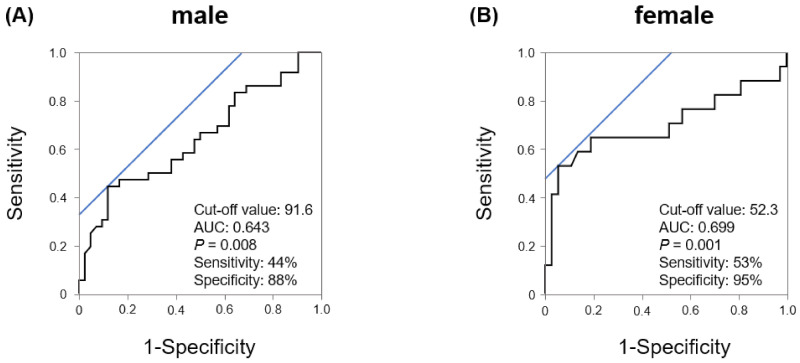
ROC curves to predict the incidence of CAS. (**A**) ROC curves for the threshold values in male patients. (**B**) ROC curves for the threshold values in female patients.

**Figure 5 jcm-10-05550-f005:**
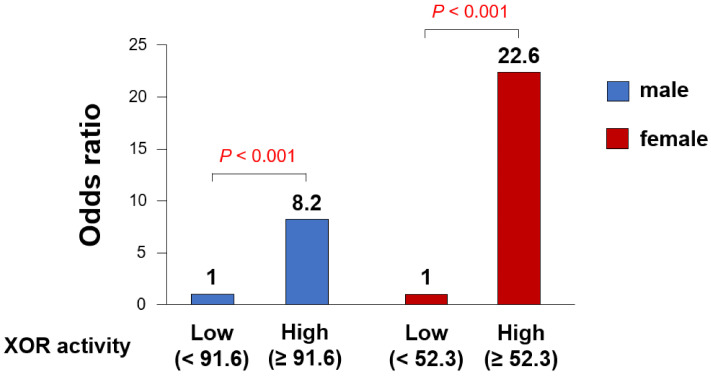
Association between plasma XOR activity and the incidence of CAS in each gender.

**Table 1 jcm-10-05550-t001:** Comparison of clinical characteristics between male and female patients.

Variables	Male *n* = 78	Female *n* = 54	*p* Value
Age (years old)	62 ± 13	68 ± 8	0.003
BMI (kg/m^2^)	23.6 ± 3.3	23.8 ± 3.9	0.728
Hypertension, *n* (%)	50 (64)	31 (57)	0.438
Dyslipidemia, *n* (%)	32 (41)	31 (57)	0.064
Diabetes mellitus, *n* (%)	12 (15)	7 (13)	0.695
Smoking, *n* (%)	43 (55)	18 (33)	0.013
Blood examination			
Triglycerides (mg/dL)	128 (93–188)	100 (76–132)	0.006
LDL-C (mg/dL)	102 ± 28	107 ± 26	0.239
HDL-C (mg/dL)	50 ± 9	62 ± 18	<0.001
HbA1c (%)	5.7 ± 0.8	5.7 ± 0.6	0.729
eGFR (mL/min/1.73 m^2^)	79 ± 22	72 ± 17	0.045
UA (mg/dL)	6.1 ± 1.3	4.7 ± 1.1	<0.001
XOR (pmol/h/mL)	51.7 (34.7–101.8)	30.3 (22.8–42.7)	<0.001
hs-CRP (mg/dL)	0.053 (0.021–0.133)	0.032 (0.018–0.087)	0.052
Medications			
ACEIs and/or ARBs, *n* (%)	37 (47)	18 (33)	0.104
CCBs, *n* (%)	52 (67)	40 (74)	0.360
Statins, *n* (%)	33 (42)	24 (44)	0.808
Antiplatelet drugs, *n* (%)	41 (53)	25 (46)	0.479
Nitrates, *n* (%)	27 (35)	12 (22)	0.121
Nicorandils, *n* (%)	27 (35)	15 (28)	0.446

Data are expressed as mean ± SD, number (percentage), or median (interquartile range). ACEIs, angiotensin-converting enzyme inhibitors; ARBs, angiotensin II receptor blockers; BMI, body mass index; CCBs, calcium-channel blockers; eGFR, estimated glomerular filtration rate; HbA1c, hemoglobin A1c; HDL-C, high-density lipoprotein cholesterol; hs-CRP, high-sensitivity C-reactive protein; LDL-C, low-density lipoprotein cholesterol; UA, uric acid; XOR, xanthine oxidoreductase.

**Table 2 jcm-10-05550-t002:** Univariate and multivariate logistic regression analysis for predicting the incidence of CAS in male patients.

	Univariate			Multivariate		
Variables	OR	95% CI	*p*-Value	OR	95% CI	*p*-Value
Age ^†^	0.903	0.570–1.418	0.656			
BMI ^†^	1.236	0.778–2.008	0.371			
Hypertension	1.231	0.486–3.167	0.662			
Dyslipidemia	2.000	0.806–5.076	0.135			
Diabetes mellitus	0.333	0.069–1.231	0.102			
Smoking	0.680	0.274–1.666	0.399			
Triglycerides ^†^	1.480	0.931–2.515	0.099			
LDL-C ^†^	1.215	0.775–1.931	0.396			
HDL-C ^†^	0.642	0.384–1.024	0.063	0.495	0.264–0.849	0.010
HbA1c ^†^	0.799	0.473–1.263	0.344			
eGFR ^†^	0.940	0.589–1.478	0.788			
UA ^†^	0.886	0.557–1.390	0.596			
XOR ^†^	2.125	1.194–4.286	0.008	2.821	1.426–6.616	0.001
hs-CRP ^†^	1.654	0.997–3.246	0.052	1.742	1.012–3.523	0.049

BMI, body mass index; CAS, coronary artery spasm; CI, confidence interval; eGFR, estimated glomerular filtration rate; HbA1c, hemoglobin A1c; HDL-C, high-density lipoprotein cholesterol; hs-CRP, high-sensitivity C-reactive protein; LDL-C, low-density lipoprotein cholesterol; UA, uric acid; XOR, xanthine oxidoreductase. ^†^ Per 1-SD increase.

**Table 3 jcm-10-05550-t003:** Univariate and multivariate logistic regression analysis for predicting the incidence of CAS in female patients.

	Univariate			Multivariate		
Variables	OR	95% CI	*p* Value	OR	95% CI	*p* Value
Age ^†^	1.745	0.945–3.570	0.076	1.742	0.989–5.522	0.054
BMI ^†^	0.886	0.485–1.598	0.687			
Hypertension	0.952	0.305–3.018	0.933			
Dyslipidemia	1.336	0.428–4.351	0.620			
Diabetes mellitus	0.304	0.015–1.987	0.236			
Smoking	2.160	0.660–7.140	0.201	3.493	0.880–15.151	0.075
Triglycerides ^†^	1.155	0.638–2.047	0.620			
LDL-C ^†^	1.144	0.634–2.050	0.646			
HDL-C ^†^	0.797	0.421–1.430	0.452			
HbA1c ^†^	0.977	0.521–1.728	0.939			
eGFR ^†^	0.967	0.527–1.725	0.910			
UA ^†^	1.416	0.801–2.598	0.232			
XOR ^†^	6.365	1.613–54.975	0.001	9.251	1.974–85.363	<0.001
hs-CRP ^†^	0.995	0.496–1.742	0.986			

BMI, body mass index; CAS, coronary artery spasm; CI, confidence interval; eGFR, estimated glomerular filtration rate; HbA1c, hemoglobin A1c; HDL-C, high-density lipoprotein cholesterol; hs-CRP, high-sensitivity C-reactive protein; LDL-C, low-density lipoprotein cholesterol; UA, uric acid; XOR, xanthine oxidoreductase. ^†^ Per 1-SD increase.

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
