# Peer review of "Gender Differences in the Impact of Plasma Xanthine Oxidoreductase Activity on Coronary Artery Spasm"

_jcm, 2021, doi:10.3390/jcm10235550_

Round 1

Reviewer 1 Report

In this study Watanabe et al. aimed to investigate gender differences in the impact of plasma xanthine oxidoreductase (XOR) activity on coronary artery spasm (CAS). The results show that (1) there was a gender difference in the distribution of plasma XOR activity; (2) the optimal cut-off values for predicting CAS were lower in women than in men; (3) high plasma XOR activity was an independent predictive factor for the incidence of CAS in both sexes; and (4) high plasma XOR activity was largely associated with the incidence of CAS in female patients than in male patients.

We thank the authors for this interesting study. The pathophysiologic hypothesis and the results are intriguing and the study design is fitting. This reviewer has some minor concerns:

  1. The methods section should be rewritten. In particular, the authors stated “we investigated plasma XOR activity in 132 patients suspected of having CAS”. They should better define the population of patients enrolled in this study. What were their symptoms? Why they underwent coronary angiography?

  1. A specific subsection on the Acetylcholine test may be valuable. The authors should discuss the protocol of Acetylcholine test they used.

  1. During the Acetylcholine test, some patients may develop a microvascular vasoconstriction/spasm, characterized by typical chest pain and/or ischemic EKG abnormalities without angiographic evidence of epicardial spasm. What is the percentage of patients enrolled in this study who experienced microvascular spasm at Acetylcholine test? What is the XOR activity in these patients? Are there any gender differences? Please, discuss.

Author Response

Response to Reviewer’s comments

We greatly appreciate the Reviewers for his or her insightful comments on our paper. The comments have helped us significantly improve the paper. Our detailed responses will follow the Reviewers comments and are shown in marked up format in the revised manuscript to facilitate the review process.

Response to the Reviewer #1

  1. The methods section should be rewritten. In particular, the authors stated “we investigated plasma XOR activity in 132 patients suspected of having CAS”. They should better define the population of patients enrolled in this study. What were their symptoms? Why they underwent coronary angiography?

[Author’s response]

Thank you for your helpful comments. According to your suggestion, we added the following sentence in page 2, line 50-52.

“We investigated plasma XOR activity in 132 patients (male, n = 78; female, n = 54) suspected of having CAS due to episodes of chest pain that occurred during rest, not exertion, in the early morning or late at night.”

  1. A specific subsection on the Acetylcholine test may be valuable. The authors should discuss the protocol of Acetylcholine test they used.

[Author’s response]

Thank you for your comments on this point. According to your suggestion, we added the following sentence in page 2, line 55-58.

“Before performing acetylcholine test, control coronary angiography was obtained. Acetylcholine was injected into the right coronary artery at dose of 20 or 50 µg and into the left coronary artery at dose of 20, 50 or 100 µg each over a period of 20 seconds.”

  1. During the Acetylcholine test, some patients may develop a microvascular vasoconstriction/spasm, characterized by typical chest pain and/or ischemic EKG abnormalities without angiographic evidence of epicardial spasm. What is the percentage of patients enrolled in this study who experienced microvascular spasm at Acetylcholine test? What is the XOR activity in these patients? Are there any gender differences? Please, discuss.

[Author’s response]

Thank you for your comments on this point. According to your suggestion, we added following sentence in page 3, line 134-137.

“Among patients without provoked CAS, there were 4 patients (male, n = 3; female, n = 1) with typical chest pain and/or ischemic electrocardiogram changes, who might develop a coronary microvascular spasm. Three male patients and 1 female patient had low XOR activity according to the ROC curve analysis, respectively.”

Reviewer 2 Report

In the present manuscript by Katanabe et al. titled »Gender differences in the impact of plasma xanthine oxidoreductase activity on coronary artery spasm« the authors aim to to inves- 41 tigate gender differences in the impact of plasma XOR activity on CAS. The manuscript represents an observational study of patients suspected of having coronary artery spasm.  

The manuscript is well written and provides insight into the relationship between XOR activity and coronary artery spasm in both genders. I believe that the manuscript should be accepted with some revision. I have provided some suggestions on ways to improve the manuscript, which are provided in the comments below.

Comments regarding revision:

  • The language in general is well understood but some grammatical and typing errors are present, which should be corrected
  • The introduction into the topic is very short albeit concise. I suggest the authors expand the introduction. Some suggestions on expanding the manuscript are: how many people with ACS have CAS and not obstructive CAD, what are the gender differences in the etiology of ACS, etc.
  • In the methods section the authors described the population that was included and possible exclusion criteria that was present. I suggest they add the timeframe in which the patients were included (e.g. during 2016) and in what center the procedure was done.
  • The information on the XOR activity assay is well presented in the methods section, as well as how data was collected from medical records etc. However, no information is available on other biochemistry tests that were reported (was the same sample used as for the XOR activity assay?) and how the eGFR was calculated (which equation was used?).
  • In the results section the authors report all results from statistical analysis that was performed. In the text they state that male subjects were older, however data in table 1 shows otherwise (males 62, females 68). I suggest they check the text for possible errors again and correct them.
  • The authors discuss all findings in the discussion and provide a concise overview of the problem. I suggest they end the discussion with some possible future use of the presented data and possible future research that should be carried out to enhance our understanding of XOR activity in CAS even more.

Author Response

Response to Reviewer’s comments

We greatly appreciate the Reviewers for his or her insightful comments on our paper. The comments have helped us significantly improve the paper. Our detailed responses will follow the Reviewers comments and are shown in marked up format in the revised manuscript to facilitate the review process.

Response to the Reviewer #2

  1. The language in general is well understood but some grammatical and typing errors are present, which should be corrected.

[Author’s response]

Thank you for your comments on this point. According to your suggestion, we corrected sentence as follows in page 10, line 198-199.

“Adipose tissue is one of the major sources of XOR, which is particularly enhanced in visceral fat in obesity [24].”

  1. The introduction into the topic is very short albeit concise. I suggest the authors expand the introduction. Some suggestions on expanding the manuscript are: how many people with ACS have CAS and not obstructive CAD, what are the gender differences in the etiology of ACS, etc.

[Author’s response]

Thank you for your comments on this point. According to your suggestion, we added the following sentence in page 1, line 30-35.

“Coronary artery spasm (CAS) is an important cause of acute coronary syndrome (ACS) and sudden death [1]. Patients with CAS is associated with poor prognosis compared with those without CAS in ACS patients [2]. There has been reported that women have higher mortality rates than men after myocardial infarction [3]. It was reported that female patients with CAS had more frequently diffuse spasm by acetylcholine tests than male patients [4].”

  1. In the methods section the authors described the population that was included and possible exclusion criteria that was present. I suggest they add the timeframe in which the patients were included (e.g. during 2016) and in what center the procedure was done.

[Author’s response]

Thank you for your comments on this point. According to your suggestion, we added the following sentence in page 2, line 52-53.

“All patients underwent an intracoronary acetylcholine provocation test in our hospital between June 2008 and October 2016.”

  1. The information on the XOR activity assay is well presented in the methods section, as well as how data was collected from medical records etc. However, no information is available on other biochemistry tests that were reported (was the same sample used as for the XOR activity assay?) and how the eGFR was calculated (which equation was used?).

[Author’s response]

Thank you for your helpful comments on this point. According to your suggestion, we added the following sentences in page 2, line 76-78.

“Other biochemistry parameters were measured using routine laboratory methods. The estimated glomerular filtration rate (GFR) was calculated by using the Japanese equation, as previously reported [17].”

  1. In the results section the authors report all results from statistical analysis that was performed. In the text they state that male subjects were older, however data in table 1 shows otherwise (males 62, females 68). I suggest they check the text for possible errors again and correct them.

[Author’s response]

Thank you for your comments on this point. According to your comments, we corrected the sentence as follows in page 3, line 103-105.

“As seen from the table, male patients were significantly younger, had higher rates of smoking, and higher levels of triglycerides and lower levels of high-density lipoprotein cholesterol (HDL-C) than the female patients.”

  1. The authors discuss all findings in the discussion and provide a concise overview of the problem. I suggest they end the discussion with some possible future use of the presented data and possible future research that should be carried out to enhance our understanding of XOR activity in CAS even more.

[Author’s response]

Thank you for your helpful comments. According to your suggestion, we added the following sentences in page 10, line 221-223.

“Decreasing XOR activity could be a novel therapeutic target for CAS, especially in female patients. Further studies are needed to examine whether XOR inhibitors are effective for the treatment of CAS.”